# Effect of concentric exercise-induced fatigue on proprioception, motor control and performance of the upper limb in handball players

Stelios Hadjisavvas[1]*, Michalis A. Efstathiou[1], Irene-Chrysovalanto Themistocleous[1], Katerina Daskalaki[2], Paraskevi Malliou[1,2], Christoforos D. Giannaki[1], Jeremy Lewis[1,3,4,5], Manos Stefanakis[1]

1 School of Health and Life Sciences, University of Nicosia, Nicosia, Cyprus, 2 Department of Physical Education and Sport Science, Democritus University of Thrace, Thrace Greece, 3 Consultant Physiotherapist, Therapy Department, Central London Community Healthcare National Health Service Trust, Finchley Memorial Hospital, London, United Kingdom, 4 Professor of Musculoskeletal Research, School of Health Sciences, University of Nottingham, Nottingham, United Kingdom, 5 Professor (Adjunct) of Musculoskeletal Research, Clinical Therapies, University of Limerick, Limerick, Ireland

* hadjisavvas.s@unic.ac.cy

**Data Availability Statement:** The data that supports the findings of this study are available in

## Abstract

### Background

The last phases of a competitive game are when shoulder injuries most commonly happen, and fatigue is thought to be a major contributing factor, perhaps because of reduced proprioception and motor control. The purpose of this study was to investigate the effect of concentric fatigue on proprioception, motor control, and performance of the upper limb in handball players.

### Methods

Forty-six right-handed handball players (all males, age 26.1 ± 5.54 years) were included in this test-retest laboratory experiment. Proprioception was assessed using joint reposition sense (JRS), threshold to detection of passive movement (TTDPM), muscle onset latency (MOL), motor control using Y balance test upper quarter (YBT-UQ) and performance using the athletic shoulder test (ASH) before and immediately after fatigue intervention. The fatigue protocol consisted of concentric, maximal effort, isokinetic contractions at $90^0$/sec with sets of 30 repetitions of the shoulder external and internal rotator muscles. Fatigue was determined by a 40% decline in the peak torque over three consecutive contractions despite reinforcing feedback and encouragement.

### Results

A significant increase in absolute angular error (AAE) was observed in all target angles in both external rotation (ER) and internal rotation (IR) directions (p < 0.01). In addition, there was a significant increase in TTDPM in internal rotation after fatigue intervention (p = 0.020.

the Mendeley Data (Digital Commons Data Repository) at https://data.mendeley.com/datasets/hrnyw4g89c/1

**Funding:** The author(s) received no specific funding for this work.

**Competing interests:** The authors have declared that no competing interests exist

**Abbreviations:** JRS, joint reposition sense; TTDPM, threshold to detection of passive movement; MOL, muscle onset latency; YBT-UQ, Y balance test upper quarter; ASH, athletic shoulder test; AAE, absolute angular error; RAE, relative angular error; ER, external rotation; IR, internal rotation; ±, standard deviation; CI, confidence interval; CS, composite score. I,Y and T: hand positions during the ASH test; PD, posterior deltoid; INF, infraspinatus..

Variable changes were found in YBT-UQ and ASH tests. Specifically, statistically significant differences were found in anteromedial (AM) (p = 0.041), superolateral (SL) reach directions (p = 0.005), composite score (p = 0.009) in the right hand and inferolateral (IL) reach direction in the left hand (p = 0.020) in the YBT-UQ. In addition, there was a significant reduction in isometric strength (ASH test) in the I position of the right hand (p = 0.010) and all positions of the left hand (p<0.05). Furthermore, there was an increase in MOL scores after fatigue but the increase was not significant (p > 0.05).

## Conclusions

Concentric fatigue of the rotator cuff muscles induces notable deficits in joint position awareness, kinesthesia, motor control, and performance of the upper extremity in elite male handball players. Although fatigue reduces reflex reaction time the effect is only marginal.

## 1. Introduction

Shoulder pain ranks as the third most common musculoskeletal problem observed in general medical practice, following low back pain and knee pain [1], and it is even more prevalent in athletic populations [2]. For instance, the prevalence of shoulder pain in handball players is estimated to be between 19% to 36% before the start of the season, and 28% during the season [3, 4]. Furthermore, a significant proportion of handball players (48%) experiencing ongoing shoulder pain are unable to participate in games or training due to increased pain intensity [5]. Dirx et al. [6] reported an increased occurrence of handball injuries in the latter half of matches, with the majority (57%) occurring in the final third of play [7] The observed increase in injuries during the final third of both practice sessions and games is postulated to be linked to alterations in neuromuscular control and proprioception induced by fatigue [8–10].

Proprioception is a fundamental aspect of neuromuscular control [11] and it has a significant impact on muscular coordination, enabling precise movement and stability of the joints [12]. Proprioception refers to the sensory input originating from peripheral regions of the body, such as the mechanical and dynamic constraints around the shoulder. This input plays a role in maintaining joint stability, controlling posture, and regulating motor functions [13, 14]. Neuromuscular control is the unconscious motor efferent response to sensory information received from proprioceptive sources [11]. The neuromuscular control of the shoulder joint involves the synchronized activation of muscles during functional activities, the simultaneous activation of shoulder muscles, the response of muscles to reflexes, and the modulation of muscle tone and stiffness [15]. Muscle forces play a crucial role in aligning the humeral head with the glenoid, while also maintaining a significant range of motion [16].

Muscle fatigue can lead to a decrease in proprioception due to a reduction in the sensitivity of muscle spindles induced by the buildup of metabolites. This buildup has the potential to interfere with the transmission of information through the muscle spindles [17]. Additionally, there may be changes in the central processing of proprioception through group III and IV afferents [18], as well as effects on the efferent pathways [19]. Muscle fatigue refers to the gradual decline in the maximum force and power that muscles can generate [20]. Fatigue is commonly categorized as either central or peripheral [21]. Central fatigue refers to a decline in the intentional activation of muscles, which is caused by a decrease in the frequency and coordination of motoneurons, as well as diminished stimulation from the motor cortex. Peripheral

fatigue refers to the reduction in the ability of muscle fibers to contract due to impaired transmission of muscle action potentials [22].

Research shows that various conditions, including pain [23], inflammation [24], injury [25, 26], and edema [26] can influence proprioception and neuromuscular control. In contrast, research demonstrates equivocal findings when it comes to fatigue, with some studies demonstrating impaired joint position sense [27–29], kinesthesia [30], and dynamic shoulder stability [31]. However, other studies have failed to observe a substantial disparity in shoulder proprioception pre and post muscle fatigue [32–34]. Furthermore, studies so far have examined the shoulder joint repositioning sense (JRS), the threshold to detection of passive movement (TTDPM) and the shoulder joint stability. No study has examined the effect of fatigue on muscle onset latency (MOL), motor control and shoulder performance. Furthermore, none of the aforementioned investigations included a high-risk population for shoulder injuries [35]. This is the first study investigating handball players with high risk for shoulder injuries from muscle fatigue.

The primary objective of this study was to examine the impact of concentric fatigue on the proprioception and motor control of the shoulder in handball players. Primary hypothesis is that concentric exercise-induced fatigue of the rotator cuff has a significant impact on the proprioception, motor control and performance of the upper limb in elite male handball players. The null hypothesis is that concentric exercise-induced fatigue has no significant effect on all outcome measures in elite male handball players.

## 2. Methods

### 2.1 Subjects

Forty-six right-handed male handball players (age 26.1 ± 5.54 yrs; height 180 ± 5.5 cm; weight 84.9 ± 13.3; body mass index: 26.2 ± 3.9 kg/m$^2$; handball experience 11.8 ± 4.7 yrs) (Table 1) from the first and second division of the Cyprus Handball Federation participated in this study. The optimal number of participants was determined using specialized software (G*power 1.3.9.4). With a significance coefficient of 0.05 and a moderate correlation between repeated measures (before and after fatigue) with an effect size of 0.3, a sample size of 30 individuals yields a power of 82% or 0.82. Therefore, even accounting for a 10% attrition rate of participants during the study, the number of 46 individuals was deemed sufficient. The participants in the study were physically active professional handball athletes who engaged in a minimum of three training sessions per week and regularly competed in official matches. Exclusion criteria were a recent shoulder injury within the past six months, previous shoulder surgery, neurological disorders, congenital stiffness, cervical radiculopathy, shoulder dislocation, intra-articular fracture, rotator cuff tear, and labral tear. In addition, every individual who reported discomfort in more than 2 clinical tests or isometric contractions during the initial evaluation was eliminated from the study and instructed to consult a physician for a definitive musculoskeletal diagnosis. The study received ethics approval by the local (national) bioethics committee (EEBK/EΠ/2020/40). The study was conducted from June 6th, 2022 until July 29th, 2022, following its approval.

Table 1. Demographic characteristics of the participants (mean ± standard deviation).

| N | Gender | Age (years) | Height (cm) | Weight (kg) | BMI (kg/m$^2$) | Experience (years) | Hand dominance |
|---|---|---|---|---|---|---|---|
| 46 | Male | 26.1 ± 5.54 | 180 ± 5.53 | 84.9 ± 13.3 | 26.2 ± 3.95 | 11.8 ± 4.79 | Right |

Abbreviations: N (number), BMI (body mass index)

## 2.2 Experimental protocol

The investigation utilized a pre-test and post-test design. Handball players were contacted via telephone. Permission to explain the study was requested, and those agreeing, the aims, procedures, commitments, and impact on time were explained. Those providing verbal consent were invited to the laboratory of the Physiotherapy School at the University of Nicosia, Cyprus, where the testing occurred. The main researcher assessed the participants to determine if they met the specific criteria for inclusion in the study. If subjects met the study participation criteria, they were provided with an informed consent document, and those that agreed to participate were requested to sign the form. Anthropometric measures were conducted with an electronic scale (Tanita WB 3000) to evaluate weight, and a statiometer (Seca, Hamburg, Germany) to determine height. Participants were instructed to wear minimal attire, excluding socks and shoes. Subsequently, the main investigator, who has gained qualifications and whose clinical practice is focused on sports and musculoskeletal physiotherapy, conducted a shoulder screening assessment. The screening examination included assessment of active and passive shoulder movements, manual isometric muscle tests (5sec duration, 'break' test) and orthopedic special tests (Neer's test, Hawkins-kennedy test, infraspinatus test, belly press test, lift off test, speed test, Yergason test, Load and shift test, apprehension test, sulcus sign, Crank test, anterior slide test) [36]. Participants who reported discomfort in two or more (or does this mean three or more) as more than 2 = 3 and above more than 2 clinical tests or isometric contractions were excluded from the study and recommended to consult a physician for a definitive musculoskeletal diagnosis. An isokinetic dynamometer (Humac Norm, CSMI solutions, USA) was utilized to measure joint repositioning sense (JRS), threshold to detection of passive movement (TTDPM), and muscle onset latency (MOL). Research has demonstrated that the isokinetic dynamometer is the most reliable tool for evaluating shoulder proprioception. Its intra-session reliability coefficient is $0.92 \pm 0.08$, as reported by Ager et al. [37]. Moreover, the isokinetic dynamometer was used to induce and quantify muscle fatigue. The Y balance–upper quarter (YBT-UQ) test and the athletic shoulder (ASH) test was used for the assessment of motor control and performance respectively [38, 39]. All outcome measures were assessed before and immediately after the acute fatigue intervention.

## 2.3 Proprioception tests

**2.3.1 Joint repositioning sense (JRS).** The JRS was assessed while each participant was in a standing position on the isokinetic dynamometer. The shoulder was positioned at 30˚ of abduction, with 0˚ of external rotation, in the plane of the scapula (30˚ anterior to the frontal plane). The elbow was flexed at 90˚, in a neutral position, and secured to the arm device (Fig 1A). The height of the dynamometer and the length of the arm device were personalized for each participant to align the axis of rotation of the dynamometer with the axis of rotation of the shoulder joint (reference point the acromion). The personalized test position was recorded and reproduced in the post-fatigue measurements. This setup allowed movement at the shoulder joint to be isolated to internal and external rotation along the shaft of the humerus. To minimize visual and auditory stimuli, the participants wore soft masks (Fig 1) that completely covered their eyes and earplugs. Prior to each measurement, every participant completed two sets of 15 repetitions of shoulder internal and external rotation exercises using the dynamometer at an angular speed of 90 degrees per second as a warm-up activity. Subsequently, six shoulder repositioning angles were measured for each participant, comprising three angles in internal rotation and three angles in external rotation of the shoulder. The six target angles that were chosen for all participants were as follows: 1) 15˚ external rotation, 2) 30˚ external rotation, 3) 45˚ external rotation, 4) 15˚ internal rotation, 5) 30˚ internal rotation,

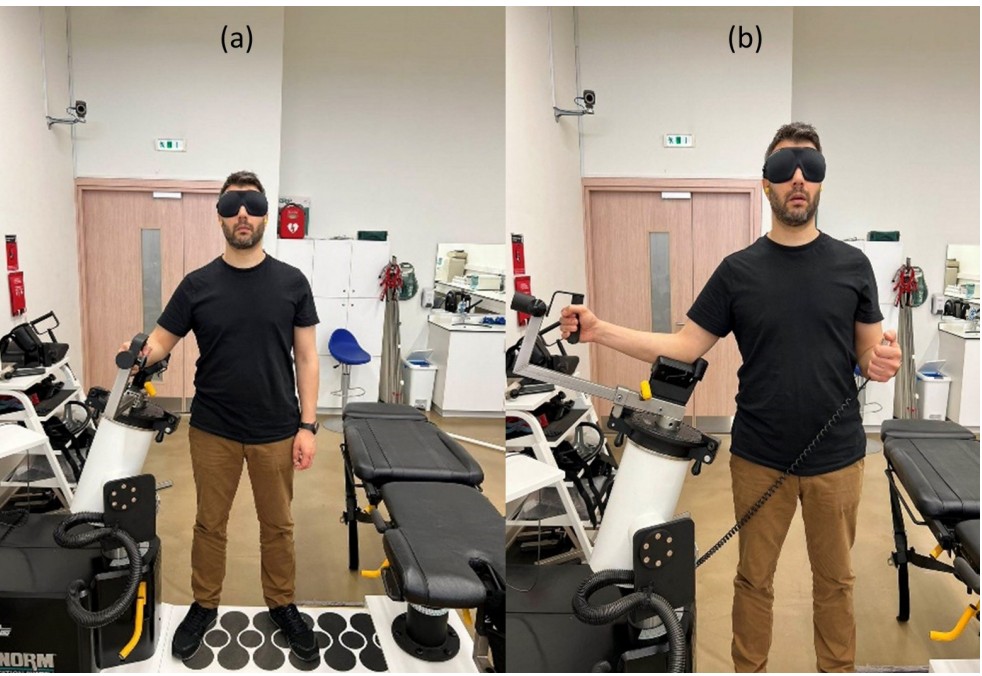

**Fig 1.** (a): The initial assessment position of the participant on the isokinetic dynamometer; (b): Assessment of TTDPM on the isokinetic dynamometer.

6) 45˚ internal rotation. The examiner positioned the shoulder at each target angle, maintaining this posture for 3 seconds, and then restored the participant's shoulder to its initial position in a passive manner. The participant was instructed to accurately reproduce the desired angle and sustain it for a duration of 3 seconds before reverting to the initial position. Three seconds was measured using a stopwatch.

Each participant conducted three trials for each target angle, and the average value was recorded. The JRS was assessed using two types of errors: the absolute angular error (AAE), which is the absolute difference between the test position and the position reproduced by the subject, and the relative angular error (RAE), which is the signed arithmetic difference between the test and response position.

**2.3.2 Threshold to detection of passive movement (TTDPM).** With the subject positioned in the exact same position as in the JRS on the isokinetic dynamometer, the visual and auditory stimuli were removed. The participant's shoulder was passively moved into internal rotation by the isokinetic dynamometer at an angular velocity of 0.5˚/s. The selection of this speed is based on the belief that it optimally activates slow-adapting joint mechanoreceptors while minimizing the activation of muscle receptors [27, 40]. Participants were directed to press a button to halt the movement when they perceived the initiation of shoulder internal rotation (see Fig 1B). Every participant conducted three trials and the average value was recorded.

**2.3.3 Muscle onset latency (MOL).** The infraspinatus and posterior deltoid MOLs were performed using surface electromyographic (EMG) electrodes during a sudden free fall of the forearm/isokinetic arm unit. Two Delsys Trigno lab EMG sensors were used to measure EMG activity. Another Delsys Trigno Lab sensor with a built-in triaxial accelerometer that was firmly attached to the dynamometer was used to detect the movement of the dynameter's arm. To ensure optimal contact of the EMG sensors and maximal signal-to-noise ratio, the skin was

shaved with a razor and was cleaned with alcohol. The sensors contain bipolar, gel-free electrodes with a diameter of 10 mm and communicate wirelessly with the computer using Delsys software (DELSYS EMGworks Analysis). The infraspinatus sensor (Fig 2) was placed two fingertips below the scapular spine and the posterior deltoid sensor (Fig 2) was placed 2 fingertips below the rear corner of the acromion as described by Cram [41]. Both sensors were stabilized in place using double sided adhesive tape [41]. Then the EMG signal of the examined muscles at rest and in maximum contraction was assessed, to identify any errors in the placement of the electrodes. Subsequently, each participant was placed on the isokinetic dynamometer with eyes and ears closed at 30˚ of shoulder abduction, 60˚ of external rotation, in the plane of the scapula (30˚ anterior to the frontal plane), with the elbow flexed at 90˚, in mid-pronation and was strapped to the arm device. While recording for a random number of seconds between 3–5, the arm of the dynamometer was dropped sharply, while measuring the start of the movement with the accelerometer and the muscle reaction with the EMG. The MOL was calculated as the difference between the onset of movement of the isokinetic arm and the onset of muscle activation (Fig 3). Muscle activation was considered to begin when the SEMG amplitude increased 3 standard deviations above the mean of the resting amplitude (Fig 5). The assessment of the infraspinatus and posterior deltoid MOL was performed before and after fatigue intervention (Fig 3).

## 2.4 Motor control and performance tests

**2.4.1 Y balance test–Upper quarter (YBT-UQ).** The YBT-UQ is a comprehensive assessment that necessitates physical strength, flexibility, neuromuscular coordination, balance, stability, and range of motion. The experiment utilized a YBT kit™, comprising of three interconnected tubular plastic bars that were labeled with markings at half centimeter intervals. Each bar has a moveable indicator plate, which the subject moves by pushing with their

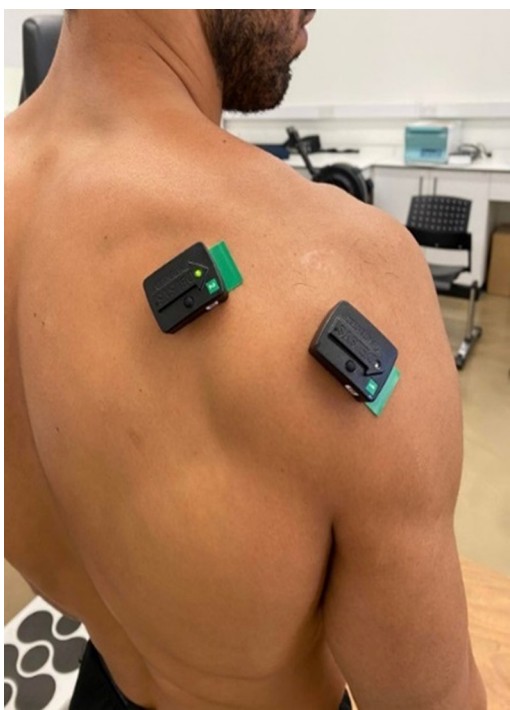

**Fig 2. The EMG electrodes placement for the infraspinatus muscle and posterior deltoid muscle.**

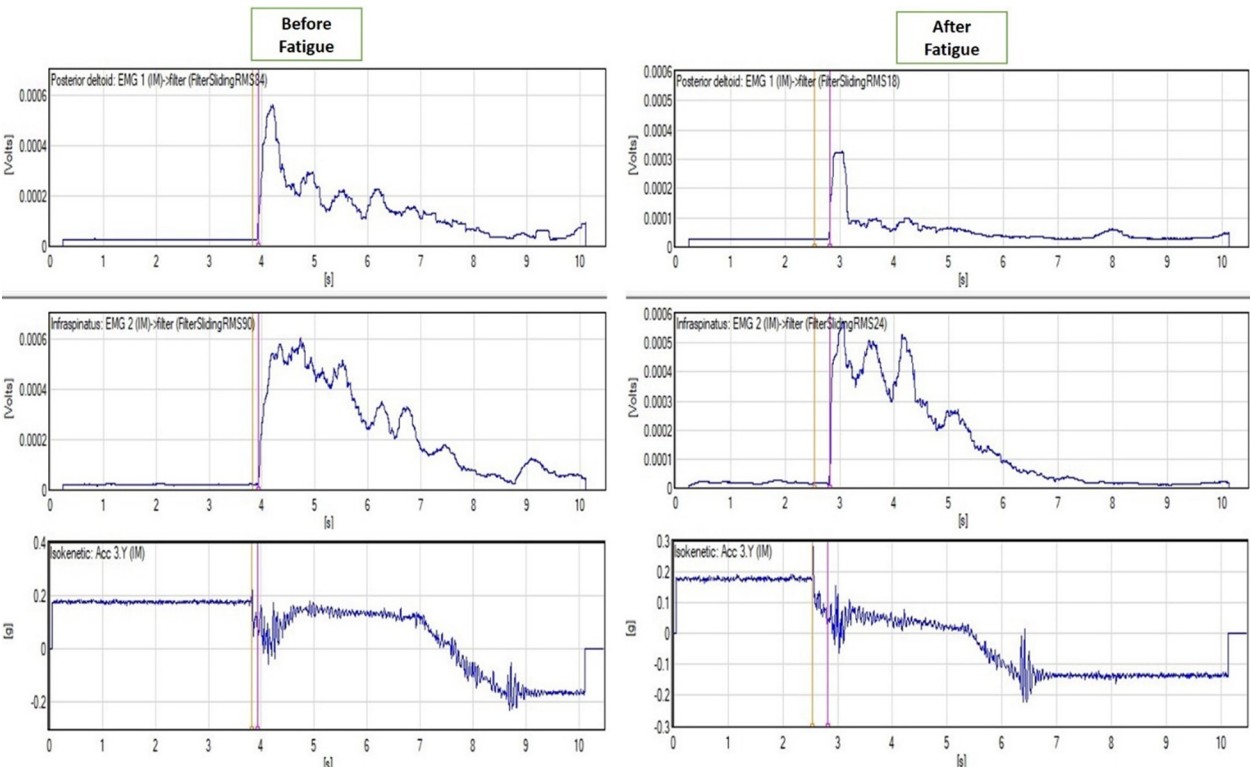

**Fig 3.** MOL before (left panel) and after fatigue (right panel). EMG of the posterior deltoid (top graphs) and infraspinatus (middle graphs). Movement of the isokinetic arm (bottom graphs) was marked with an orange vertical line. The onset of muscle activation was marked with a purple vertical line. The time difference between these lines represents MOL. (DELSYS EMGworks Analysis).

hand/fingers. The test was conducted by initiating the subject in a push-up stance with their feet positioned at a distance equal to the breadth of their shoulders. The objective of the test was for the participant to maintain a push-up position on the central platform of the YBT device and move the indicator plate in the anteromedial (AM), superolateral (SL), and infero-medial (IL) directions (Fig 4) using the other hand. Following a warm-up test consisting of three attempts in each direction, three additional attempts were conducted in each direction. The maximum reach distances in each direction were added together to obtain a composite reach distance, which was then adjusted to account for the length of the upper limb. This facili-tated an analysis of the overall proficiency demonstrated on the assessment. Prior to the test, the length of the subject's upper limb, from the 7th cervical vertebrae to the tip of the middle finger, was measured using a tape measure. In cases where the subject made more than four unsuccessful attempts, a score of zero was documented for that trial. A participant's attempt was deemed a failure if they engaged in any of the following actions: throwing instead of push-ing the box, returning to the beginning push-up posture without maintaining control, making contact with the floor using their hand prior to pushing the box, or having their feet lose con-tact with the ground. All individuals successfully completed the trials during the test without any failures. The YBT-UQ test [38, 42] demonstrated high reliability (ICC 0.80–1.0) and showed no disparity in performance between the dominant and non-dominant limbs. These findings suggest that the YBT-UQ is a dependable instrument for evaluating sports perfor-mance during the recovery process of shoulder, upper extremity, and spine injuries.

**2.4.2 The athletic shoulder test (ASH test).** The ASH test was developed to evaluate the isometric strength of the shoulder. The test was conducted according to the methodology

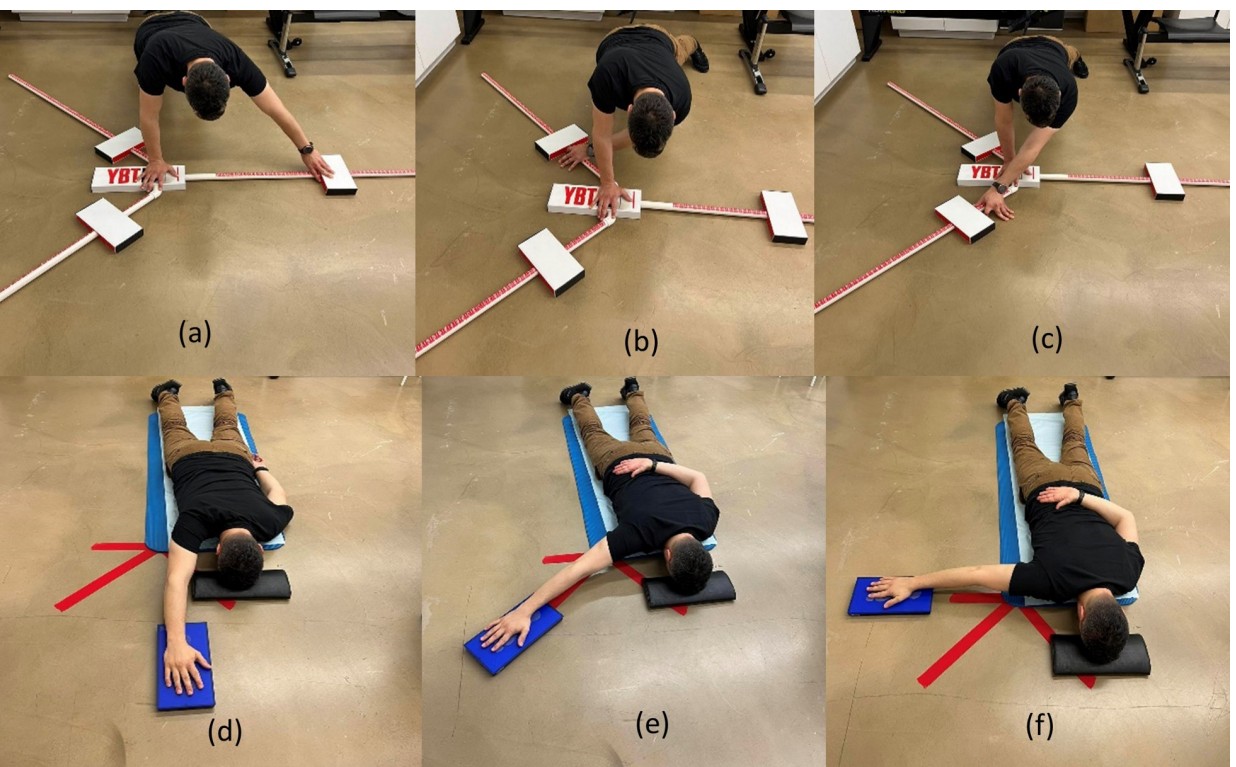

**Fig 4.** The YBT-UQ (a to c) and the ASH test (figures d to f); (a): anteromedial (AM) reach; (b): the inferolateral (IL) reach; (c): the superolateral (SL) reach; (d): the "I" position; (e): the "Y" position; (f) the "T" position.

outlined by Ashworth et al. [39], with participants lying face down on the floor and their foreheads touching a 4 cm elevated cushion. The hand's palm was positioned atop a force plate. The participant's hand in this test was positioned in three configurations ("I", "Y", and "T") (Fig 4). When in the "I" position, the shoulder is fully abducted (aligned with the body), the forearm is in pronation, and the "heel" of the hand serves as the primary point of contact with the force platform. The arm was positioned at an abduction angle of 135˚ in the "Y" position and 90˚ in the "T" position. The elbow was required to be fully extended in all tests. The arm on the other side was positioned behind the back to prevent the elbow from being anchored to the floor and to offer support against trunk rotation during the "Y" and "T" tests. However, during the "I" test, the arm was kept at the participant's side because the trunk experienced lesser twisting forces. Subjects were instructed to exert maximum force on a portable uniaxial force plate (K-Force Plates, Kinvent, Montpellier, France) for a duration of three seconds at each test position. Data on force was gathered at a frequency of 300 Hz. The maximum force was obtained for each test and adjusted to bodyweight. Prior research [39] has indicated that the ASH test exhibits exceptional reliability across several test positions, as proven by an inter-day reliability coefficient (ICC) ranging from 0.94 to 0.98. Furthermore, the test exhibited significant absolute reliability values (SEM 4.8–10.8) and the inter-day measurement error was below 10% in all test locations (CV 5.0–9.9%), except for the non-dominant arm I-position (CV 11.3%). The minimum detectable change ranged from 13.2 to 25.9 N [39].

## Reduction in ER and IR torque due to fatigue

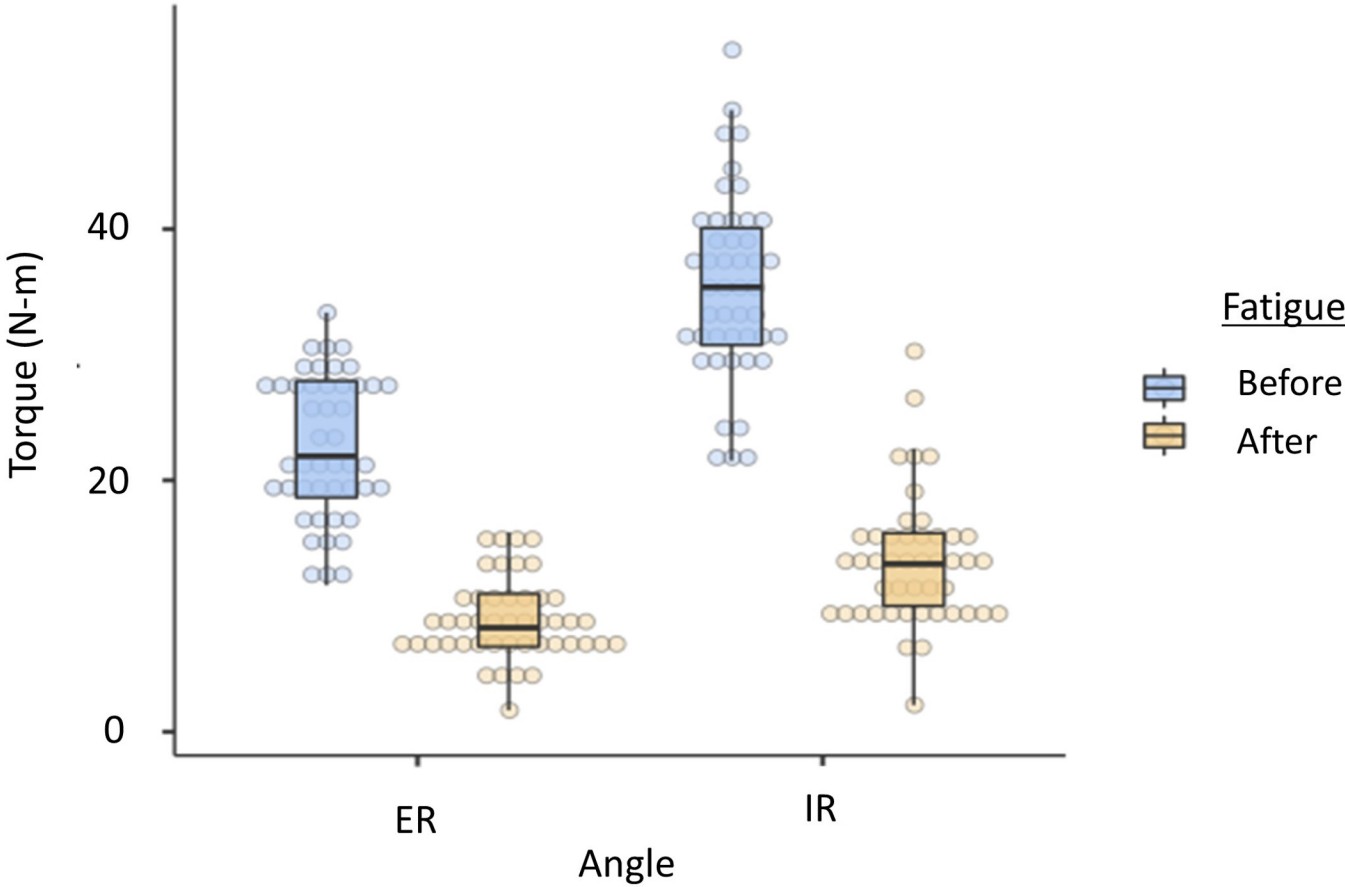

**Fig 5. The shoulder external rotation (ER) and internal rotation (IR) before and after fatigue intervention.**

### 2.5 Exercise-induced fatigue protocol

Each participant was placed in the standing position on the isokinetic dynamometer. The right shoulder was placed at 30˚ of abduction, 0˚ of external rotation, in the plane of the scapula (30˚ anterior to the frontal plane), with the elbow flexed at 90˚, in mid-pronation and was strapped to the arm of the device. Each participant actively performed maximal voluntary, concentric, isokinetic contractions from 60˚ of shoulder internal rotation to 60˚ of shoulder external rotation to induce concentric muscle fatigue of the shoulder external and internal rotator muscles. This test was performed in the same position as the one used to induce fatigue. Initially the examiner instructed each participant to perform 10 submaximal (20% of maximal voluntary contraction) concentric contractions with an angular speed 90 degrees per second to familiarize themselves with the specific movement. Next, peak isokinetic torque of internal and external rotators was recorded by asking the subject to perform 10 repetitions with maximal effort. This value was the reference value for the induction of fatigue. After a 5 minutes rest, each participant performed sets of 30 maximal effort concentric isokinetic contractions with 90 degrees per second angular speed of the external and internal rotator muscles. The criterion to consider muscle fatigue was the drop of the isokinetic peak torque to 40%

of the reference value (60% reduction) and remained below this value, irrespective of positive reinforcement, for 3 consecutive repetitions. If the subject exceeded the reference value of peak torque in the first set, then the reference value was instantly updated, and the same rule was used for the induction of fatigue. The sets needed to induce fatigue varied between one and five, most commonly three. The same protocol was used in a previous study [29]. After the test, muscle fatigue was assessed by the percentage decrease in peak torque between the first ten and the last ten repetitions (mean peak torque of first ten repetitions - mean peak torque of last ten repetitions / mean peak torque of first ten repetitions) during the thirty repetitions sets of isokinetic exercise at 90 deg/sec.

## 2.6 Statistical analysis

Jamovi (Version 2.3.26) was used for statistical analysis. Descriptive statistics were used to calculate means and standard deviations (SDs) of the collected variables. Shapiro-Wilk test was used to assess data for normality prior to the analysis. The paired samples t-test was used to compare the mean differences before and immediately after the fatigue intervention for the absolute angle error (AAE), the relative angle error (RAE), the threshold to detect passive movement (TTDPM), the MOL, YBT-UQ and the ASH test scores. For the non-normally distributed variables, the Wilcoxon test was used for pre-post fatigue intervention comparisons. Statistical significance was set at $p < 0.05$ and the 95% CI was calculated for all mean differences.

## 3. Results

The peak torque of shoulder internal rotation and shoulder external rotation decreased by 61.90% (Fig 5) and 60.92% (Fig 5) correspondingly after muscle fatigue, as indicated by the fatigue index. The mean value of external rotation peak torque during the initial ten and final ten repetitions was recorded as 33.4 N (± 5.65) and 15.9 N (± 3.27), respectively. The mean internal rotation peak torque during the initial and final 10 repetitions was 54.3 N (± 7.32) and 30.3 N (± 5.16) respectively.

Following fatigue intervention, a significant increase in absolute angular error was observed in all target angles in both ER and IR directions (p<0.01) (Table 2). A similar significant

**Table 2. JRS AAE and RAE scores pre- and post- fatigue intervention.**

| Type of error | Target angles | Pre-Fatigue | Post Fatigue | p value | Mean Difference (95% CI) |
|---|---|---|---|---|---|
| AAE | ER15˚ | 1.91 ± 0.957 | 4.72 ± 1.976 | < .001 | -2.81 (-3.500, -2.122) |
| AAE | ER30˚ | 2.64 ± 1.208 | 5.35 ± 2.487 | < .001 | -2.71 (-3.440, -1.979) |
| AAE | ER45˚ | 2.69 ± 1.360 | 5.91 ± 3.327 | < .001 | -3.22 (-4.250, -2.187) |
| AAE | IR15˚ | 1.97 ± 1.006 | 3.11 ± 1.481 | < .001 | -1.15 (-1.580, -0.707) |
| AAE | IR30˚ | 2.60 ± 1.254 | 4.69 ± 2.429 | < .001 | -2.09 (-2.750, -1.426) |
| AAE | IR45˚ | 2.17 ± 0.902 | 3.51 ± 2.028 | < .001 | -1.50 (-2.340, -0.670) |
| RAE | ER15˚ | -0.06 ± 1.230 | 3.68 ± 3.230 | < .001 | -3.76 (-4.720, -2.794) |
| RAE | ER30˚ | 1.20 ± 2.130 | 3.59 ± 4.340 | < .001 | -2.39 (-3.600, -1.185) |
| RAE | ER45˚ | 0.05 ± 1.990 | 4.56 ± 4.600 | < .001 | -4.51 (-5.810, -3.205) |
| RAE | IR15˚ | -0.09 ± 1.430 | 1.85 ± 2.390 | < .001 | -1.95 (-2.700, -1.197) |
| RAE | IR30˚ | 0.34 ± 2.480 | 3.33 ± 3.760 | < .001 | -2.99 (-4.080, -1.900) |
| RAE | IR45˚ | -0.15 ± 1.750 | 0.86 ± 3.770 | 0.101 | -1.03 (-2.270, 0.210) |

Abbreviations: JRS, joint repositioning sense; AAE, absolute angular error; RAE, relative angular error; ER, external rotation; IR, internal rotation; ±, standard deviation; CI, confidence interval.

**Table 3. TTDPM scores pre and post- fatigue intervention.**

| Movement | Pre-Fatigue | Post Fatigue | p value | Mean Difference (95% CI) |
|---|---|---|---|---|
| IR | 5.91 ± 3.640 | 7.39 ± 6.290 | 0.020 | -1.50 (-2.500, -8.630) |

Abbreviations: TTDPM, threshold to detection of passive movement; IR, internal rotation; ±, standard deviation; CI, confidence interval.

increase in relative angular error was observed in five out of six target angles after muscle fatigue (p<0.01). No significant difference was observed in IR45˚ (p = 0.101) (Table 2). In addition, there was a significant increase in TTDM in internal rotation after fatigue intervention (p = 0.020) (Table 3).

Regarding the YBT-UQ, statistically significant differences were found in anteromedial (AM) (p = 0.041) and superolateral (SL) reach directions (p = 0.005) in the right hand and in inferolateral (IL) reach direction in the left hand (p = 0.020) (Table 4). Furthermore, the composite score demonstrated a statistically significant difference only in the right hand (p = 0.009) (Table 4). There was a significant reduction of isometric strength in the I position of the right hand (p = 0.010) and to all positions of the left hand (p<0.05) in the ASH test (Table 5). Following muscular fatigue, there was an elevation in the reaction time (MOL) of the posterior deltoid and infraspinatus muscle (Table 5). Nevertheless, this rise was not statistically significant (p>0.05).

## 4. Discussion

The main aim of this study was to examine the impact of concentric muscle fatigue on the proprioception and motor control of the shoulder rotator cuff in handball players. The results of the present study confirmed the primary hypothesis that concentric fatigue negatively influences all outcome measures assessed. The significant findings of the present study indicate that concentric fatigue of the rotator cuff muscles induces notable deficits in proprioception, motor control, and performance. In addition, no significant differences were found in the reflex activation of the muscles after a sudden perturbation due to fatigue.

The peak incidence of injury is observed during the last third of training and competition in both soccer [43] and handball [6, 7]. These changes are believed to be associated with alterations in proprioception and neuromuscular control because of fatigue [8, 10]. Furthermore, research has demonstrated that fatigue may have a detrimental impact on athletes' performance in several functional assessments, including proprioception tests [44–46], balance tests [47, 48], and muscle strength tests [49], making athletes more susceptible to injury.

**Table 4. The YBT-UQ mean (±) standard deviation scores pre and post- fatigue intervention.**

| Hand Position | Pre Fatigue | Post Fatigue | p value | Mean Difference (95% CI) |
|---|---|---|---|---|
| AM_Right | 94.5 ± 10.950 | 92.8 ± 11.050 | 0.041 | 2.50 (1.190, 4.500) |
| IL_Right | 83.4 ± 12.590 | 80.9 ± 12.280 | 0.073 | 2.47 (-0.240, 5.200) |
| SL_Right | 65.1 ± 10.530 | 62.2 ± 12.870 | 0.005 | 2.93 (0.942, 4.930) |
| AM_Left | 92.6 ± 11.130 | 93.2 ± 10.800 | 0.428 | -0.50 (-2.500, 1.00) |
| IL_Left | 83.5 ± 9.640 | 80.8 ± 11.160 | 0.020 | 2.73 (0.462, 5.020) |
| SL_Left | 67.0 ± 11.040 | 65.6 ± 11.700 | 0.114 | 1.43 (-0.356, 3.230) |
| CS right | 87.3 ± 11.400 | 84.7 ± 11.600 | 0.009 | 2.56 (0.681, 4.440) |
| CS left | 87.3 ± 10.600 | 86.0 ± 10.800 | 0.110 | 1.28 (-0.301, 2.870) |

Abbreviations: YBT-UQ, Y balance test upper quarter; AM, anteromedial; IL, inferolateral; SL, superolateral; CI, confidence interval; CS, composite score.

**Table 5. ASH test and MOL scores pre- and post- fatigue intervention.**

| Variable | Pre-Fatigue | Post Fatigue | p value | Mean Difference (95% CI) |
|---|---|---|---|---|
| I_Right | 13.02 ± 3.490 | 11.86 ± 3.330 | 0.010 | 1.15 (0.287, 2.023) |
| Y_Right | 10.59 ± 2.450 | 10.10 ± 2.480 | 0.107 | 0.49 (-0.110, 1.100) |
| T_Right | 9.42 ± 2.010 | 9.05 ± 2.030 | 0.103 | 0.37 (-0.078, 0.828) |
| I_Left | 12.70 ± 3.320 | 11.27 ± 3.040 | < .001 | 1.54 (1.095, 1.975) |
| Y_Left | 9.85 ± 2.270 | 9.07 ± 2.280 | 0.002 | 0.77 (0.304, 1.238) |
| T_Left | 9.21 ± 1.970 | 8.47 ± 2.130 | 0.005 | 0.73 (0.238, 1.234) |
| PD MOL | 0.179 ± 0.137 | 0.262 ± 0.218 | 0.088 | -0.0391 (-0.1018, 0.00780) |
| INF MOL | 0.180 ± 0.138 | 0.267 ± 0.220 | 0.092 | -0.0392 (-0.0960, 0.00586) |

Abbreviations: ASH, the Athletic Shoulder Test; MOL, muscle onset latency; ±, standard deviation; CI, confidence interval; I,Y and T, hand positions during the ASH test; PD, posterior deltoid; INF, infraspinatus.

The results of the present study suggest that after the induction of fatigue, JRS is significantly disturbed. These findings provide evidence for the "dysfunctional mechanoreceptor" idea, which suggests that muscular fatigue might decrease the sensitivity of the muscle spindle, leading to notable disruptions in proprioception [32]. This notion has received support from other investigations [27, 29]. The cause of this malfunction is not completely understood. Fatigue-induced alterations in local muscle metabolism may be a potential explanation for this impairment [50]. Pedersen et al. [50] found that elevated levels of lactic acid, creatine-kinase, bradykinin, arachidonic acid, and serotonin inside the muscles during muscle fatigue can potentially disrupt the proper functioning of muscle spindles and, consequently, affect proprioception. Due to the greater intensity of local blood flow and metabolic changes, muscle mechanoreceptors are more susceptible to impact compared to articular receptors [50].

The findings of the current study align in part with the results of previous research [27–29, 31, 51]. Lee et al. [27] discovered that with the onset of muscular fatigue, there was a notable and statistically significant rise in position error specifically during the JRS test for shoulder external rotation, but no such increase was observed for shoulder internal rotation. On the contrary, Kablan et al. [28] reported that novice volleyball players made considerably more position errors in internal rotation compared to external rotation after experiencing shoulder muscular fatigue. Lida et al. [29] reported that accuracy to reproduce a position decreased in both shoulder internal and external rotation following their fatigue intervention. Similar results were found by Myers et al. [31], who reported a significant impairment of shoulder joint repositioning sense in both directions of shoulder rotation after fatigue. Furthermore, Voight et al. [32], found that shoulder muscle fatigue significantly affected the shoulder joint repositioning sense in the direction of external rotation at an angle of 75˚.

In contract, other studies did not identify any significant differences in the JRS before and after fatigue protocols [33, 34]. Spargoli [34] expected that eccentric exercise would have a stronger detrimental impact on shoulder proprioception compared to concentric exercise. However, there were no significant differences for the absolute angular inaccuracy comparing the pre- and post-fatigue measurements in either of the groups. Sterner et al [33] failed to verify that shoulder muscle fatigue had a significant negative impact on JRS during both active and passive movements.

In an attempt to account for the different findings between the studies, methodological differences need to be considered. In the current experiment, the standing position was chosen as a means to induce muscular exhaustion and assess proprioception. On the other hand, Sterner et al. [33] employed the supine position. Additionally, variations existed in the target angles for

the shoulder joint repositioning sense. The present study employed three distinct target angles (15˚, 30˚, and 45˚) to measure shoulder internal and external rotation. In contrast, Spargoli [34] used a single target angle of 30˚ of external rotation, while Sterner et al. [33] evaluated the angle in the middle range. Furthermore, there were variations in the fatigue procedure, including the angular speed of the isokinetic dynamometer and the criteria used to determine muscle fatigue. The angular velocity chosen for the fatigue method in the present study was 90 degrees per second. However, Spargoli [34] employed an angular speed of 180 deg/sec, whereas Sterner et al. [33] did not provide any information regarding the angular speed. In the present investigation, the fatigue confirmation resulted in a higher percentage drop of peak torque (60%) compared to Sterner's et al study [33] (50%). Finally, this study utilized handball players, whereas the other two investigations [33, 34] included subjects who were healthy non-athletes.

The present study found a negative association with TTDPM and shoulder muscular fatigue. This means that shoulder muscle fatigue detrimentally impacts on TTDPM and the threshold is larger after fatigue and small in the non-fatigued condition. A possible reason for this is muscle fatigue leads to decreased sensitivity of muscle receptors that are primarily responsible for kinesthetic information [52]. Prior research has verified that muscle spindles serve as the principal receptors in kinesthesia [53–55]. The findings of the current investigation are consistent with the results of Carpenter et al. [30], which they had found that the mean TTDPM (both internal and external rotation combined) increased by 73% after the fatigue intervention. Carpenter et al. [30], employed tested participants in sitting and used a higher angular velocity of 1 deg/sec on the isokinetic dynamometer, which was greater than the current study's velocity of 0.5 deg/sec. another difference is that the current study only included elite male handball players, whereas Carpenter et al. studied healthy non-athletes volunteers [30]. Despite the clear methodological differences between the two studies (current and Carpenter et al.), results were identical. This increases the generalizability of the findings. On the other hand, Sterner et al. [33] did not observe that shoulder muscle fatigue had a substantial impact on TTDPM [35, 37]. The variation in the initial assessment position (supine) and the selection of a distinct sample (healthy non-athletes) used by the Sterner et al. [33] may account for the disparity observed in the outcomes.

Muscle fatigue not only impairs proprioception, but also leads to a decrease in shoulder stability and mobility [56, 57]. In this study, after muscular fatigue, the YBT-UQ test findings suggest a greater impact on superolateral reach, anteromedial reach, and composite score the right hand (superolateral reach, anteromedial reach, and composite score) which was the limb that was exercised, compared to the left hand. Fatigue is believed to decrease the sensitivity of receptors in the stabilizer muscles, reducing their ability to co-contract during closed motor chain activities. The decreased stability of the supported upper extremity leads to reduced mobility of the opposite hand [31]. The current investigation specifically induced fatigue in the right upper limb. Due to this protocol, the right side was more affected than the left side. With this finding, the effect of fatigue on the stability and mobility of the upper limb has been confirmed. The negative impact of the unilateral fatigue protocol on motor control in the exercised limb supports other studies in that had used bilateral fatigue protocols, and reported significant impairments of motor control in both upper limbs [56, 57].

Another possible explanation for the adverse impact of fatigue on motor control may stem from a deficiency in the central processing of proprioceptive signals induced by central exhaustion [58]. The role of central nervous system factors, such as changes in neurotransmitter levels or reduced motor cortex function, in the diminished voluntary activation of skeletal muscles is another possible mechanism for the reduction of motor control [59]. However, this is only hypothetical as no measures of central fatigue were used in this study.

Salo and Chaconas [56] observed a significant decrease in YBT-UQ scores for all reaching tasks in both limbs after performing bilateral exercises including shoulder press, machine row, prone push-up, and pull-up exercises. Their study utilised a distinct exhaustion methodology for weightlifters and not for handball players. In a more recent study, Bauer et al. [57] handball players experienced significantly decreased YBT-UQ scores, specifically in the SL reach direction for both upper limbs after undergoing a fatigue protocol involving push-up activities. In contrast, Bauer et al. [57] induced fatigue using push-ups. Myers et colleagues [31], conducted a study to investigate the impact of shoulder muscular fatigue on shoulder stability using the single-arm dynamic stability (SADST) test. The results showed that there were no significant differences in sway velocity between the pre-test and post-test measurements. However, it was observed that there was a notable rise in the occurrence of falls following the fatigue intervention. This led to the conclusion that fatigue can impair the co-contraction ability of the shoulder while in the closed kinetic chain position. One explanation for the difference may be attributed to differences in outcome measurements as Myers et al. [31], utilized expand here (SAD) to evaluate shoulder stability, whereas the current study used the YBT-UQ.

Another significant finding from the current study was that shoulder muscular fatigue decreased shoulder isometric strength. Specifically, the ASH test showed a significant decrease in muscle strength at the "I" position of the right hand and all positions of the left hand. No research has been undertaken to examine the impact of rotator cuff muscle fatigue on the ASH test. Muscle fatigue did not have a substantial impact on the other assessment positions of the right hand ("T", "Y"). One potential reason is that the ASH test was the final assessment conducted following the exhaustion regimen, and it is plausible that muscle strength was restored. Li et al. [60] supported this idea, concluding that the restoration of shoulder muscular strength occurred faster than the recovery of shoulder joint position sense following fatigue. The reduction in the force production of the left, non-fatigued side is difficult to interpret. It is possible that reduced stabilization due to fatigue on the opposite right side. This remains an hypothesis and its veracity requires further investigation.

The study findings suggests that fatigue in the rotator cuff muscles increases the delay in the reflex response time of the shoulder external rotator muscles (infraspinatus and posterior deltoid) in response to external disruptions. Specifically, the percentage delay of MOL for the posterior deltoid muscle was 46.36% and 48.33% for the infraspinatus muscle after fatigue intervention. Fatigue is thought to decrease muscle reflex responses via affecting intrafusal characteristics, presynaptic inhibition of Ia afferents, and intrinsic features of motoneurons [61, 62]. Although the result was not statistically significant there is no way to judge if it is clinically significant as no reference value exist for the delay of muscle onset. No research has been conducted to investigate how rotator cuff muscle fatigue affects reflex reaction time. However, the present study revealed higher percentage values of delay in MOL compared to previews studies [61–66] despite the lack of statistical significance. Prior research has investigated how lower limb muscle fatigue affects the reflex reaction time after tibial translation of the hamstring muscles. Behrens et al. [61] found that due to fatigue the percentage delay of the biceps femoris and semitendinosus reflex reaction time was 4.29% and 4.69% respectively in women. On the other hand, other researchers [62], found that the percentage delay of the biceps femoris muscle was 12.05%. Sun et al. [64], investigated the impact of fatigue (90 minutes of soccer game) on MOL of the peroneal muscle was. The results showed that the percentage delay of the peroneal muscle reflex activation time was 21.06%. Other researchers [65] revealed 14.06% delay in reflex reaction time of tibialis anterior muscle after 90 minutes of soccer game. The erector spinae is another muscle group that was investigated despite the reflex reaction time after fatigue. Zuriaga et al. [66] demonstrated 14.51% and 7.04% delay in the reflex reaction time of the erector spinae muscle at the L3 spinal level. Granata and Slota [62], investigated the

effect of fatigue on MOL of internal obligue and rectus abdominis muscle. In this case, the internal oblique muscle showed 9.75% faster reflex reaction time compared to the pre fatigue value. Nevertheless, the rectus abdominus muscle demonstrated a 46.05% delay.

Limitations in the methods of the current investigation must be considered and acknowledged. Initially, the possibility of induced central fatigue may have influenced between hand comparisons. Our study design involved adult male handball players, leaving the impact of muscle fatigue on proprioception in female and juvenile handball players, and people not playing handball unexplored. Furthermore, muscle fatigue was induced within a controlled laboratory setting using an isokinetic dynamometer. The muscle fatigue encountered by handball players during real-life conditions, such as repetitive throwing motions, sprinting, and jumping, differs from that observed in laboratory settings. These limitations need to be considered when using the results of the study.

## 5. Conclusions

Concentric exercise-induced fatigue of the rotator cuff muscles can lead to notable deficits in joint position awareness, kinesthesia, motor control, and performance of the upper extremity in elite male handball players. Moreover, fatigue might result in a decrease in reflex reaction time. These findings can assist sports health practitioners in comprehending the adverse impact that athlete fatigue can exert on the neuromuscular system. Hence, it is essential to assess proprioception, motor control, and performance both before and after fatigue in order to detect any dysfunctions that may arise as a result of fatigue. To improve proprioception, motor control and performance, it is recommended to engage in targeted exercises while fatigued in order to heighten the sensitivity of mechanoreceptors in fatigue situations.

## Acknowledgments

Mrs Georgia Christodoulou (Head librarian) for her help in the search process.

## Author Contributions

**Investigation:** Stelios Hadjisavvas.

**Methodology:** Stelios Hadjisavvas, Manos Stefanakis.

**Supervision:** Manos Stefanakis.

**Writing – original draft:** Stelios Hadjisavvas.

**Writing – review & editing:** Michalis A. Efstathiou, Irene-Chrysovalanto Themistocleous, Katerina Daskalaki, Paraskevi Malliou, Christoforos D. Giannaki, Jeremy Lewis, Manos Stefanakis.

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
