## [Decision Letter · Decision Letter 0]

2 Oct 2024

PONE-D-24-22537Effect of concentric exercise-induced fatigue on proprioception, motor control and performance of the upper limb in handball playersPLOS ONE

Dear Dr. Hadjisavvas,

Thank you for submitting your manuscript to PLOS ONE. After careful consideration, we feel that it has merit but does not fully meet PLOS ONE’s publication criteria as it currently stands. Therefore, we invite you to submit a revised version of the manuscript that addresses the points raised during the review process.

We look forward to receiving your revised manuscript.

Kind regards,

Nili Steinberg

Academic Editor

PLOS ONE

Journal Requirements: When submitting your revision, we need you to address these additional requirements. 1. Please ensure that your manuscript meets PLOS ONE's style requirements, including those for file naming. The PLOS ONE style templates can be found at https://journals.plos.org/plosone/s/file?id=wjVg/PLOSOne_formatting_sample_main_body.pdf and https://journals.plos.org/plosone/s/file?id=ba62/PLOSOne_formatting_sample_title_authors_affiliations.pdf 2. In the online submission form, you indicated that "The datasets generated and/or analysed during the current study are available from the corresponding author on reasonable request." All PLOS journals now require all data underlying the findings described in their manuscript to be freely available to other researchers, either 1. In a public repository, 2. Within the manuscript itself, or 3. Uploaded as supplementary information.This policy applies to all data except where public deposition would breach compliance with the protocol approved by your research ethics board. If your data cannot be made publicly available for ethical or legal reasons (e.g., public availability would compromise patient privacy), please explain your reasons on resubmission and your exemption request will be escalated for approval. 3. Please review your reference list to ensure that it is complete and correct. If you have cited papers that have been retracted, please include the rationale for doing so in the manuscript text, or remove these references and replace them with relevant current references. Any changes to the reference list should be mentioned in the rebuttal letter that accompanies your revised manuscript. If you need to cite a retracted article, indicate the article’s retracted status in the References list and also include a citation and full reference for the retraction notice.

Reviewers' comments:

Reviewer's Responses to Questions

**Comments to the Author**

1. Is the manuscript technically sound, and do the data support the conclusions?

Reviewer #1: Yes

Reviewer #2: Yes

2. Has the statistical analysis been performed appropriately and rigorously? 

Reviewer #1: Yes

Reviewer #2: Yes

3. Have the authors made all data underlying the findings in their manuscript fully available?

Reviewer #1: Yes

Reviewer #2: Yes

4. Is the manuscript presented in an intelligible fashion and written in standard English?

Reviewer #1: Yes

Reviewer #2: Yes

5. Review Comments to the Author

Reviewer #1: Introduction

1. I would recommend to move it down to the research perspective: "Research has demonstrated that various conditions, including pain [17], inflammation [18], injury [19, 20], and edema [20] can influence proprioception and neuromuscular control. In addition, fatigue is another important component that can significantly impact proprioception abilities [21]".

2. "Published research demonstrates equivoca (...)" Please add a clear perspective including gaps in knowledge and innovationss.

3. Please add hypothesis.

Discussion

1. Please provide an information about hypothesis.

2. Please explain the results supported by physiological and biomechanical mechanisms.

Reviewer #2: The title " Effect of concentric exercise-induced fatigue on proprioception, motor control and performance of the upper limb in handball players" is clear and effectively conveys the focus of the study on the effects of concentric exercise-induced fatigue on proprioception, motor control and performance of the upper limb in handball players. As a reviewer, I commend the authors for their meticulous and insightful study on the related handball specific fatigue in relation with the effect of concentric exercise-induced fatigue on proprioception, motor control and performance of the upper limb in handball players. The thoroughness and dedication demonstrated in conducting this research significantly contribute to the advancement of our understanding in this emerging field. The valuable insights provided by the authors serve as a commendable contribution to the scientific community. Thank you for your commitment to excellence in this research. In my modest opinion as a reviewer the different part of the manuscript were well presented, detailed and supported well the objective and the rational of the study.

6. PLOS authors have the option to publish the peer review history of their article (what does this mean?). If published, this will include your full peer review and any attached files.

Reviewer #1: No

Reviewer #2: **Yes: **Seifeddine Brini

---

## [Author Response · Author response to Decision Letter 0]

26 Oct 2024

I have replied to the comments in the file I have uploaded.

---

## [Editor Report · Decision Letter 1]

21 Nov 2024

Effect of concentric exercise-induced fatigue on proprioception, motor control and performance of the upper limb in handball players

PONE-D-24-22537R1

Dear Dr. Hadjisavvas,

We’re pleased to inform you that your manuscript has been judged scientifically suitable for publication and will be formally accepted for publication once it meets all outstanding technical requirements.

Kind regards,

Nili Steinberg

Academic Editor

PLOS ONE
---

## [Editor Report · Acceptance letter]

27 Nov 2024

PONE-D-24-22537R1 

PLOS ONE

Dear Dr. Hadjisavvas, 

I'm pleased to inform you that your manuscript has been deemed suitable for publication in PLOS ONE. Congratulations! Your manuscript is now being handed over to our production team.

Kind regards, 

on behalf of

Prof. Nili Steinberg 

Academic Editor

PLOS ONE